# Single-component color-tunable circularly polarized organic afterglow through chiral clusterization

Hui Li[1], Jie Gu[1], Zijie Wang[1], Juan Wang[1], Fei He[1], Ping Li[1], Ye Tao [1✉], Huanhuan Li[1], Gaozhan Xie[1], Wei Huang [1,2✉], Chao Zheng[1] & Runfeng Chen [1✉]

Circularly polarized organic afterglow (CPOA) with both long-lived room-temperature phosphorescence (RTP) and circularly polarized luminescence (CPL) is currently attracting great interest, but the development of multicolor-tunable CPOA in a single-component material remains a formidable challenge. Here, we report an efficient strategy to achieve multicolor CPOA molecules through chiral clusterization by implanting chirality center into non-conjugated organic cluster. Owing to excitation-dependent emission of clusters, highly efficient and significantly tuned CPOA emissions from blue to yellowish-green with dissymmetry factor over $2.3 \times 10^{-3}$ and lifetime up to 587 ms are observed under different excitation wavelengths. With the distinguished color-tunable CPOA, the multicolor CPL displays and visual RTP detection of ultraviolent light wavelength are successfully constructed. These results not only provide a new paradigm for realization of multicolor-tunable CPOA materials in single-component molecular systems, but also offer new opportunities for expanding the applicability of CPL and RTP materials for diversified applications.

[1] State Key Laboratory of Organic Electronics and Information Displays & Institute of Advanced Materials (IAM), Nanjing University of Posts & Telecommunications, 9 Wenyuan Road, Nanjing 210023, China. [2] Institute of Flexible Electronics (IFE), Northwestern Polytechnical University, Xi'an 710072 Shanxi, China. ✉email: iamytao@njupt.edu.cn; provost@nwpu.edu.cn; iamrfchen@njupt.edu.cn

Circularly polarized luminescent (CPL) materials with efficient emission dissymmetry factor ($g_{lum}$) have received widespread attention recently on account of their attractive potential applications in diversified optoelectronic applications[1–6]. Especially, the pleochroism of chiral luminescent materials can empower multiplexing confidentiality for multilevel data encoding and encryption[7]; the multicolor CPL materials with stimuli-response attributes are idea emissive mediums for brilliant liquid crystal and stereoscopic display[8]; high sensitivity and spatial resolution luminescence sensors for biological analytes and object identification could also be gifted by employing polychromatic chiral molecules as optical marker recognition[9]. In these applications, multicolor-tunable CPL emissions are highly desired, but can be only realized in multicomponent material systems on the basis of various design methods including multicomponent supramolecular assembly[10,11], chemical additive induced excited state modulation[12–14] and solvation effect enabled structural color change[15], etc. Nevertheless, it remains a daunting challenge to develop efficient multicolor-tunable CPL emission from a single-component material system with the capability to dynamically and selectively respond to the changes of physical parameters of external stimuli such as electrical and magnetic fields, pressure, and optical input[16].

Circularly polarized organic afterglow (CPOA) with both the ultralong-lived triplet excited state for room-temperature phosphorescence (RTP) and efficient CPL features is an amazing type of luminance that has emerged as a hot research topic recently because of the distinguished photophysical phenomena in various remarkable applications[17–20]. Over the past few years, a set of design approaches including multicomponent host–guest[21], ionic co-crystal[22], and chiral chain engineering[23] have been proposed to promote the intersystem cross (ISC) and suppress non-radiative decay of triplet excitons, and to boost efficient chirality transfer from chiral groups to chromophores simultaneously for the realization of CPOA materials with lifetime over 100 ms. Nonetheless, the afterglow emission colors of CPOA materials are limited to fixed emission colors, mostly in yellow or yellowish-green under ambient conditions[19,21–24]; it is rather difficult to develop multicolor-tunable CPOA materials due to the complicated chirality transfer between the chiral unit and the RTP emission group for CPL as well as the inefficient RTP color tuning for multicolor emission, especially in a molecule. To the best of our knowledge, the CPOA material showing multicolor-tunable CPL emission from a single-component molecule system has not yet been reported.

Here, we propose an efficient strategy to realize color-tunable CPOA from a single-component non-conjugated molecule by chiral clusterization. Specially, the non-conjugate cluster shows size-dependent clusterization-triggered emission (CTE) via through-space conjugation (TSC) for efficient excitation-dependent color-tunable organic afterglow emission due to the presence of varied luminescent centers with different cluster sizes (Fig. 1a)[25,26]. Inspired by the flexible chiral chain engineering to realize CPOA in a single-component purely organic small molecule, we suspect that the color-tunable CPOA can be also designed by implanting the chiral chain into a single-component cluster crystal. Indeed, through a large number of in-plane and interlayer molecular interactions and the dense in-plane and interlayer non-covalent bond network to support the formation of varied emission species and suppression of the non-radiative transition of excitons (Fig. 1b, c)[27–29], efficient color-tunable CPOA with emission peaks from 470 to 530 nm, lifetimes up to 587 ms, and the $|g_{lum}|$ over $2.3 \times 10^{-3}$ were observed upon tuning the excitation wavelength of ultraviolet (UV) light. Based on the spectacular feature of the color-tunable CPOA molecules, multicolor CPL displays and visual RTP detection of UV light were

successfully established. These findings in purposefully constructing multicolor CPOA with the on-demand color-tuning ability through the elaborate molecular design by chiral clusterization would provide a promising platform for the exploration of single-component smart CPL and RTP materials.

## Results

**Material design and synthesis.** To confirm our hypothesis, we designed and prepared a pair of enantiomers composed of chiral *trans*−1,2-diamidocyclohexane core and two succinic acid arms (Fig. 1d)[30]. In this motif, chiral *trans*-1,2-diamidocyclohexane can not only employ as a chiral center but also can facilitate the intersystem crossing (ISC) promoted by the nitrogen (N) atoms to accelerate the production of triplet excitons. Meanwhile, bearing the carbonyls (C=O) and intra/intermolecular bonding sites, the succinic acid unit can remarkably boost the ISC process and more importantly[31–33], establish strong non-covalent bond interactions between different molecules to form interlocked molecular system for the suppression of non-radiative decay and formation of varied clusters with different sizes. In addition, benefitting from the multiple heteroatoms and varied clusters, TSC originated from the orbital overlap of (n, π*) and (π, π*) can be effectively conferred to endow tunable triplet energy levels in different clusters[34]. As a proof of concept, with the molecular design strategy, the enantiomers of (R, R)-DAACH, and (S, S)-DAACH were facilely synthesized by a one-step amidation reaction between *trans*-(1 R, 2 R)/(1 S, 2 S)-diamidocyclohexane and succinic anhydride. This one-step reaction enables high yields up to 88% and the resultant molecules were systematically characterized by nuclear magnetic resonance (NMR) spectra, high-resolution mass spectra (HRMS), matrix-assisted laser desorption/ionization time of flight mass spectrometer (MALDI-TOF MS) and thermal physical properties (Supplementary Figs. 1–13).

**Photophysical properties.** Expectantly, excitation-dependent color-tunable CPOA is successfully realized in single-component chiral cluster crystal of (R, R)-DAACH and (S, S)-DAACH, showing long-lived emissions ranging from blue to yellowish-green with moderate $|g_{lum}|$ over $2.3 \times 10^{-3}$ under the excitation by 240, 340, and 360 nm, respectively (Fig. 2a, b and Supplementary Fig. 14). The chiral characteristics were also observed, when (R, R)-DAACH and (S, S)-DAACH powders were dispersed (50 wt%) in potassium bromide slice (Supplementary Fig. 15). The almost mirror circular dichroism signals of (S, S)-DAACH and (R, R)-DAACH ethanol solutions with strong signals at ~205 nm further confirm the successful introduction of chirality in these materials (Supplementary Fig. 16). The excitation-phosphorescence spectra of (R, R)-DAACH and (S, S)-DAACH powders demonstrated that with varying the excitation wavelength tuning from 200 to 450 nm (Fig. 2a, bottom panel and Supplementary Fig. 17), the gradually red-shifted CPOA with main emission peaks tuning from 470 to 530 nm and decreased luminescent intensities were achieved, which were also manifested by the resolved CPOA spectra (Supplementary Figs. 18-19) and corresponding Commission International de l'Eclairage (CIE) (Fig. 2c) upon excitation by a different wavelength ranging from 220 to 360 nm with a step of 20 nm; also, a nearly linear relationship between the variations in CIE coordinates and excitation wavelength was obtained, indicating the great feasibility of this color-tunable CPOA in serving as RTP sensing chart for detecting UV light.

To further investigate the influence of the excitation wavelength on the color-tunable CPOA properties, the time-resolved emission spectra were performed[35]. As shown in Fig. 2d and

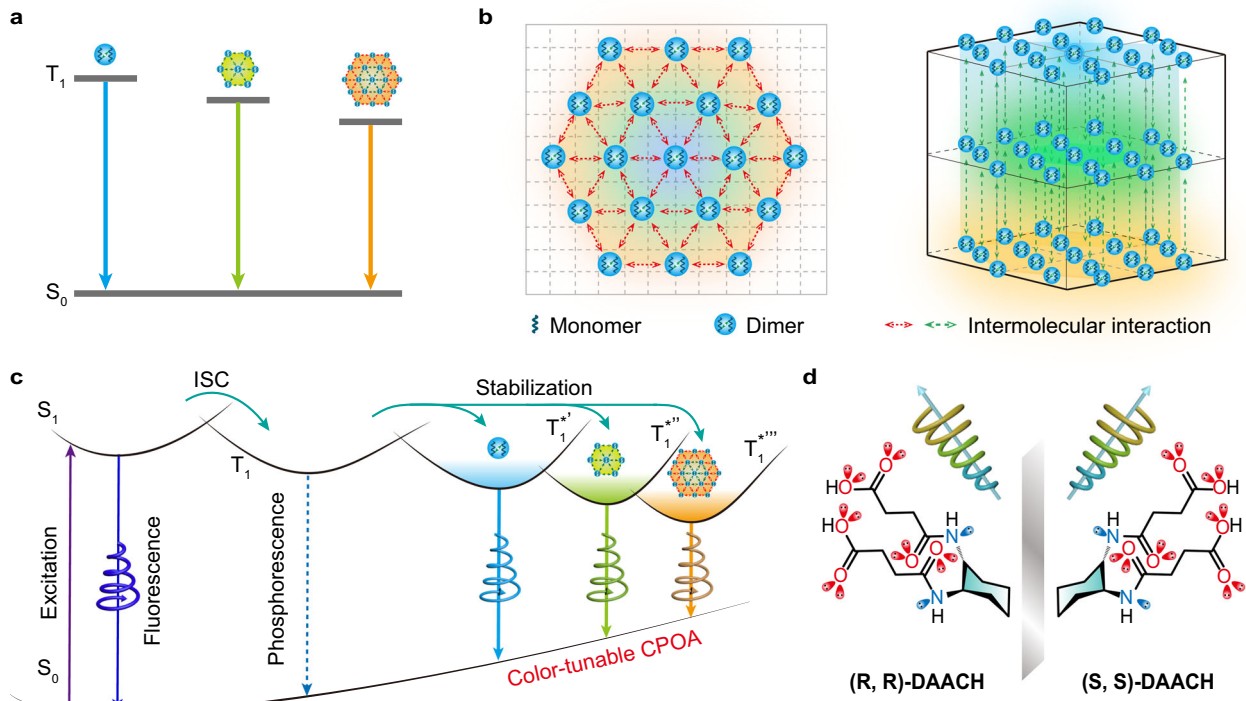

**Fig. 1 Molecular design of single-component color-tunable CPOA. a** Proposed mechanism of color-tunable phosphorescence emission from different cluster sizes showing tunable excited energy levels. **b** Top and lateral views of molecular packing in the chiral cluster crystal. The dotted arrows represent intra/intermolecular interactions. The blue ball and the hexagons with different background colors ranging from blue to orange represent size-dependent CTE centers showing color-tunable CPOA. **c**, **d** Proposed mechanism (**c**) and molecular design (**d**) of color-tunable CPOA from a single-component chiral cluster crystal. So is the ground state. $S_1$ and $T_1$ represent the lowest singlet and triplet excited states. $T_1^{*'}$, $T_1^{*''}$, and $T_1^{*'''}$ are the stabilized triplet excited states originated from varied emission species showing different cluster sizes, respectively.

Supplementary Figs. 20–21, both (R, R)-DAACH and (S, S)-DAACH powders indicated excitation-dependent CPOA decay profiles, displaying tunable ultralong lifetimes spanning from 130 to 460 ms for (R, R)-DAACH and 324 to 587 ms for (S, S)-DAACH powders upon excitation turned from 220 to 360 nm, respectively. These distinct variations in lifetime further confirmed the presence of disparate emissive species[26]. Meanwhile, the transient emission decay images of (R, R)-DAACH and (S, S)-DAACH powders verified that this color-tunable emission is quite stable over time under the excitation by different wavelengths (Fig. 2e and Supplementary Fig. 22). The changes in CPOA intensities as a function of irradiation time and intensity showed that excellent CPOA properties could be stimulated under the short excitation duration of 0.1 s and low excitation power density of 0.11 mW/cm$^2$ (Fig. 2f, g and Supplementary Figs. 23–25). Possibly due to the relatively slow and weak intersystem crossing rates, it needs considerable time to achieve the stable state of triplet excitons for afterglow emission, resulting in the slightly enhanced RTP with the increase of photo-irradiation time (Fig. 2f and Supplementary Fig. 23). Moreover, with the excitation by 240 nm, CPOA revealed higher intensity compared to that of the excitation wavelength of 360 nm. As demonstrated in Fig. 2h, with the excitation by UV light from 200 to 330 nm the sky-blue CPOA emissions from (R, R)-DAACH powder were much higher than these of yellowish-green; in contrast, when they were motivated by UV light from 330 to 450 nm, the yellowish-green CPOA became the dominated emission; these color changes upon tuning the excitation wavelength further confirmed the presence of varied emissive species for the color-tunable CPOA emission.

To gain a deeper insight into these distinct photophysical properties, the steady-state photoluminescence (SSPL) spectra were further monitored. The SSPL of (R, R)-DAACH is also excitation-dependent, exhibiting slight hypochromatic shift for varied colors from sky-blue to blue and tunable quantum yields when the excitation wavelength was changed from 260 to 360 nm (Supplementary Fig. 26 and Table 1). This blue-shifted emission should be due to the gradually decreased intensities of the CPOA emission bands originating from 470 to 550 nm as the excitation wavelength varied from 260 to 360 nm. To prove this phenomenon, the SSPL spectra at 77 K of (R, R)-DAACH powder were also tested. As shown in Supplementary Fig. 27, because the non-radiative transition of triplet emission is greatly inhibited under freezing conditions, much-enhanced CPOA emissions were observed at 77 K compared to those under room temperature; and much obviously blue-shifted emissions from green to blue were observed in (R, R)-DAACH powder, suggesting again that the abnormal hypochromatic shifted emission obtained under the excitation turning from 260 to 360 nm should be attributed to the decreased intensities of CPOA emissions.

**Mechanism insights of CPOA.** To seek the underlying origination of these unique optical properties, a further set of experimental and theoretical investigations including the SSPL in solutions, single-crystal analyses, and time-dependent density functional theory calculations on the (R, R)-DAACH were conducted[36]. As shown in Supplementary Figs. 28, 29, the excitation-dependent emissions were also observed in SSPL spectra of (R, R)-DAACH in ethanol solutions with varied

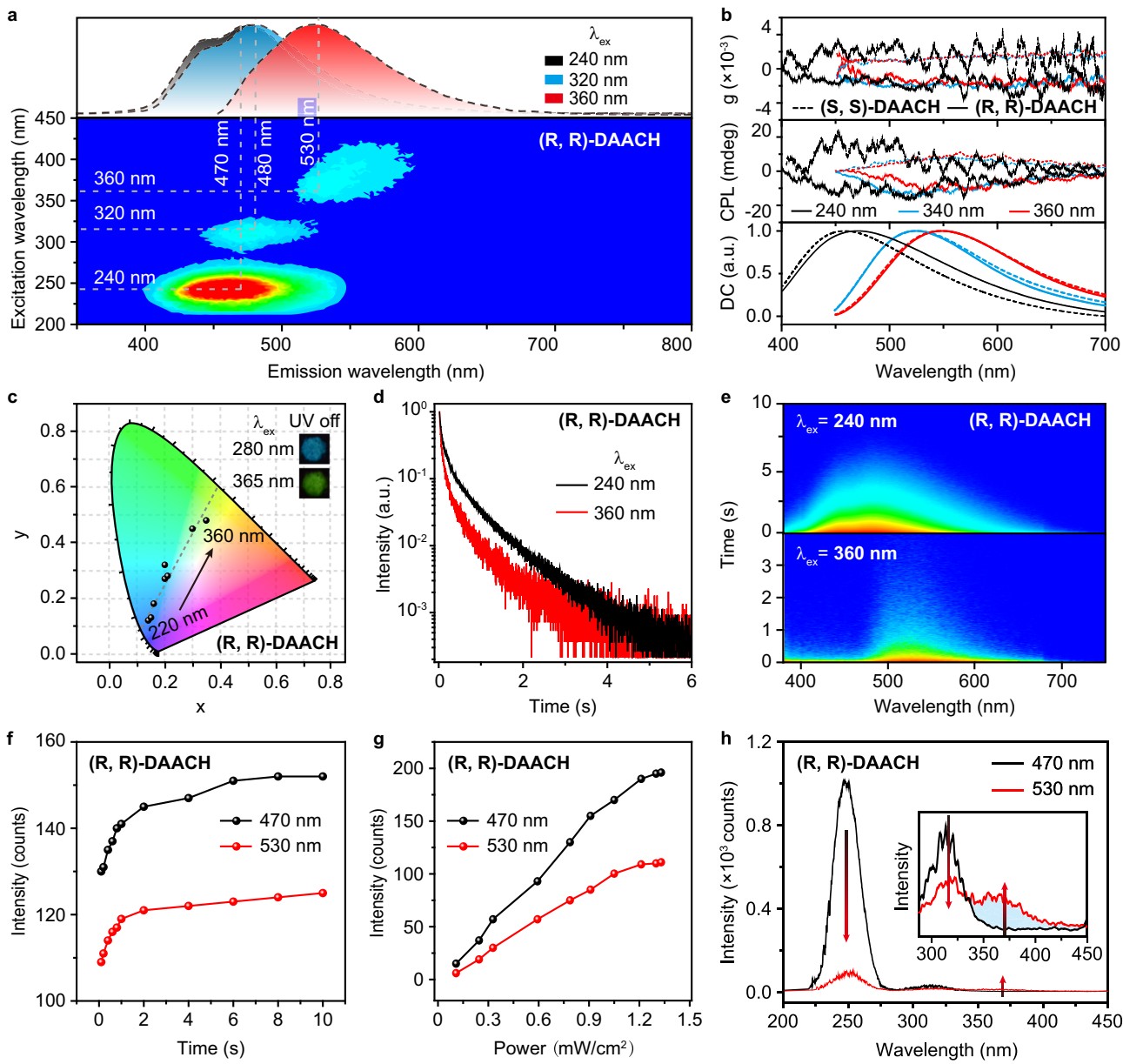

**Fig. 2 Photophysical properties of the single-component color-tunable CPOA materials under ambient conditions. a** Excitation-phosphorescence mapping of (R, R)-DAACH powder. The upper panel shows the CPOA spectra of (R, R)-DAACH powder under excitation by 240, 320, and 360 nm, respectively. **b** CPL properties of (R, R)-DAACH and (S, S)-DAACH powders under excitation by 240, 340, and 360 nm, respectively. **c** CIE coordinates variations of color-tunable CPOA spectra of (R, R)-DAACH powder upon excitation by a different wavelength ranging from 220 to 360 nm with a step of 20 nm. The insets exhibit the CPOA photographs of (R, R)-DAACH powder taken after turning off 280 and 365 nm excitation light, respectively. **d** Decay profiles of CPOA emission of (R, R)-DAACH powder upon excitation by 240 and 360 nm, respectively. **e** Transient emission decay images of (R, R)-DAACH powder following excitation by 240 (top) and 360 nm (bottom), respectively. **f, g** CPOA intensity changes of (R, R)-DAACH powder at 470 and 530 nm as a function of irradiation time (**f**) and intensity (**g**) upon excitation by 240 (black line) and 360 nm (red line), respectively. **h** Excitation spectra of (R, R)-DAACH powder by monitoring CPOA peaks of 470 and 530 nm, respectively.

concentrations upon excitation at different wavelengths under ambient conditions, which are consistent with the variation trend of (R, R)-DAACH powder (Supplementary Fig. 26), verifying again the presence of multiple emissive species as further revealed by varied fluorescence lifetimes and excitation-dependent cryogenic phosphorescence spectra under the excitation by different wavelengths (Supplementary Figs. 30–31). Notably, as the concentration increased from $10^{-5}$ to $10^{-2}$ M, the well-resolved peaks in the range of 400–500 nm exhibited largely enhanced intensities with slightly decreased lifetimes (Supplementary Fig. 32) and the red-shifted absorption spectra were found

(Supplementary Fig. 33). This high concentration enhanced luminescent intensities and red-shifted absorption bands of these unconventional luminophores should be ascribed to the enhanced molecular p-π conjugation and intense intermolecular interactions in the aggregated state, thus leading to the efficient electron delocalization on diverse emissive centers via TSC for the formation of effective CTE[34,37].

**Crystallography analyses**. To reveal the presence of different clusters in CPOA molecules, the intermolecular interactions and

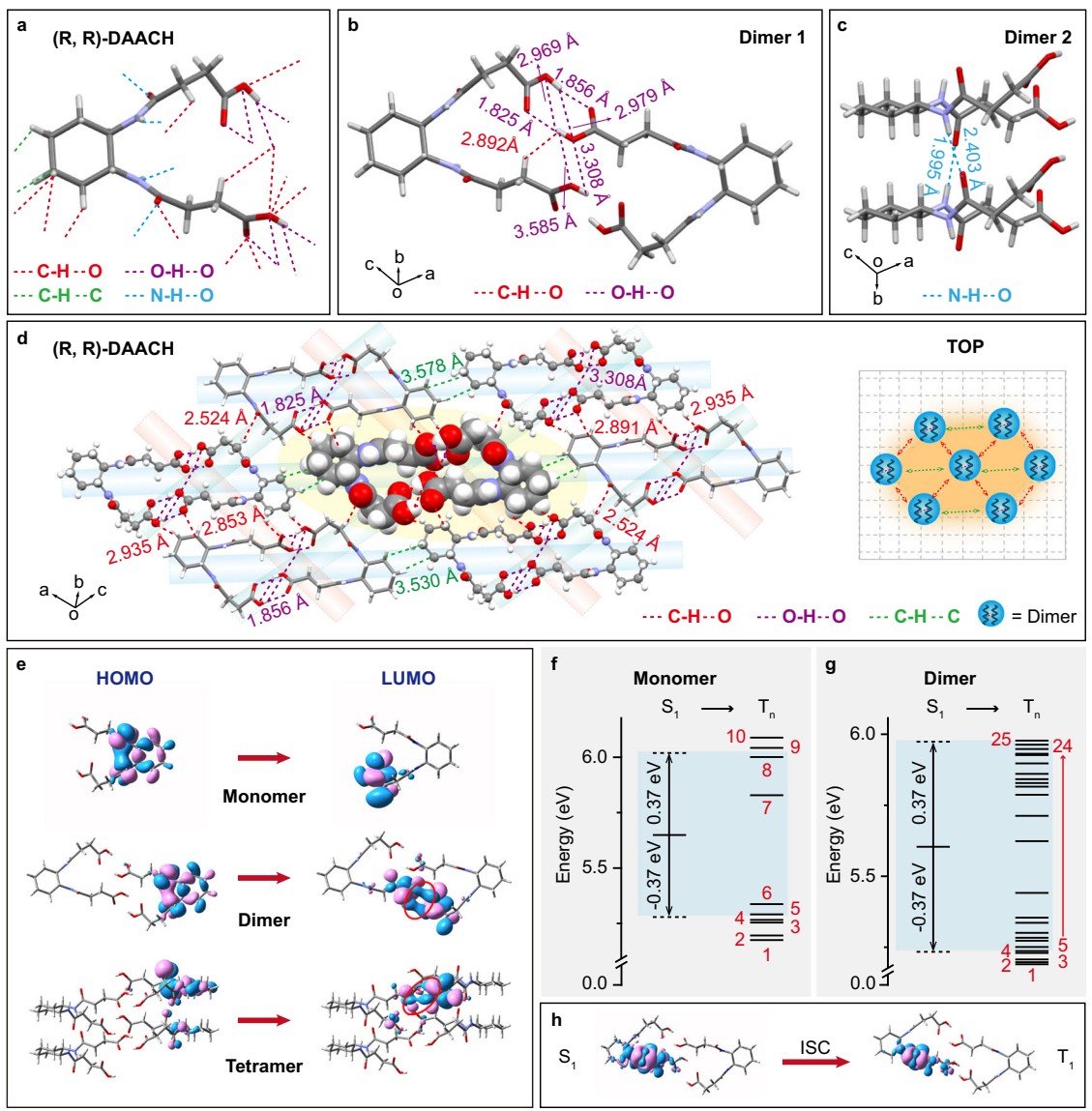

**Fig. 3 Single crystal and theoretical analyses of color-tunable CPOA materials. a–d** Intermolecular interactions of monomer (**a**), dimer (**b**, **c**), and molecular packing (**d**) in (R, R)-DAACH single crystal. The central dimer (highlighted by space fill mode) is stabilized by six identical dimers via efficient intermolecular interactions through neighbor dimers, thus leading to the diversified emission species with different cluster sizes. **e** Frontier molecular orbital distributions for the selected monomer, dimer, and tetramer in (R, R)-DAACH single crystal. **f**, **g** TD-DFT calculated energy level diagram of selected monomer (**f**) and dimer (**g**) in (R, R)-DAACH single crystal. **h** Electron density difference in the singlet and triplet excited state of the dimer.

packing modes in the single crystal were carefully analyzed. (R, R)-DAACH adopted hairpin-shaped conformation in crystal (Fig. 3a), showing multiple effective intra/intermolecular interactions including C-H•••O, O-H•••O, C-H•••C, and N-H•••O. And obvious intermolecular interactions of O-H•••O with distances ranging from 1.825 to 3.585 Å, C-H•••O with a distance of 2.892 Å, C-H•••C with distances of 3.530 and 3.578 Å, and N-H•••O with distances of 1.995 and 2.403 Å were observed in different dimers and multimers (Fig. 3b, c and Supplementary Figs. 34, 35), which could not only promote the formation of rigid molecular configuration for suppressing non-radiative transitions but also facilitate TSC among different molecules[38,39]. These intense intermolecular interactions were also theoretically identified by the independent gradient model (IGM) analysis[40], exhibiting a large IGM isosurface between adjacent molecules in both dimers, trimers, and tetramers (Supplementary Fig. 36). Extendedly, (R, R)-DAACH exhibited various interlocked dimers and multimers in the entire crystal (Fig. 3c and Supplementary

Fig. 34) by utilizing the efficient intra/intermolecular interactions in neighboring dimers; and the central dimer was bonded by adjacent six dimers (Fig. 3d) in the same plane and these bonded dimers were further stabilized through intensive interlayer intermolecular interactions of N-H•••O (Supplementary Fig. 35). These confinements in dimers could gift the different emission species with regulated and enhanced TSC for color-tunable CPOA that responds to varied excitation wavelength, probably leading to the central dimer for sky-blue afterglow, bonded dimers for green, and extendedly interlocked dimers for yellowish-green afterglows[34,40].

**Theoretical investigations**. To further manifest our conceptions, we performed the theoretical calculations of the monomer and selected aggregates in (R, R)-DAACH single crystal. From the simulation results, the highest occupied molecular orbital (HOMO) and the lowest unoccupied molecular orbital (LUMO)

(Fig. 3e) were separately localized on cyclohexane and acid arms in monomer; importantly, the HOMOs and LUMOs are remarkably delocalized on the cyclohexane and acid arms among different monomers for efficient through-space electron delocalization in the aggregated state. Furthermore, with the increment of the aggregate size, more obvious through-space delocalization and decreased energy bandgaps were observed (Supplementary Fig. 37). This kind of spatial frontier molecular orbital delocalization can not only narrow the energy bandgap but also lead to different clusters with plentiful triplet energy levels; these findings can be experimentally revealed by the red-shifted emission bands upon tuning the excitation wavelength from 240 to 360 nm and theoretically verified by the gradually decreased excited energy levels with the increasing of the aggregated size of a cluster. Moreover, to theoretically investigate the effective ISC in the aggregated state, the calculations of time-dependent density functional theory (TD-DFT) were carried out[32,41]. From the simulated excited state energy levels, there are only five triplet states showing singlet and triplet splitting energy ($\Delta E_{ST}$) lower than 0.37 eV for enabling possible ISC in (R, R)-DAACH monomer according to the energy gap law (Fig. 3f and Supplementary Table 3)[23]. In contrast, the total number of plausible ISC channels in the aggregated state was significantly increased to 20 for dimer, 30 for trimer, and 37 for tetramer (Fig. 3g, Supplementary Fig. 38, and Tables 4–8), respectively, thus facilitating high-efficiency ISC for the promotion of triplet excitons (Fig. 3h). Accordingly, on the basis of the experimental and theoretical investigations, we reason that the strong intra/intermolecular interactions in the same plane and interlayer should be responsible for the formation of different clusters to render efficient TSC for achieving varied emission species with tunable excited energy levels; on account of boosted ISC in the aggregated state, the photo-motivated singlet excitons can be facilely transformed to triplet excitons; taken together, on account of different emission species and high ISC rate in the aggregated state, excitation-dependent color-tunable CPOA can be successfully achieved in this unconventional chirality cluster crystal (Fig. 4a).

**Universality of the strategy.** To testify the universality of this chiral clusterization strategy in constructing single-component color-tunable CPOA materials, other two pairs of enantiomers of (R, R)/(S, S)-DAPCH and L/D-serine were explored. Indeed, (R, R)/(S, S)-DAPCH demonstrate superb excitation-dependent CPOA properties, showing tunable emission color ranging from 410 to 530 nm (Supplementary Fig. 39). And, the single-component color-tunable CPOA was also achieved in L/D-serine, exhibiting tunable chiral afterglow emissions spanning from deep blue to green (Supplementary Fig. 40).

**Applications of the CPOA molecule.** Benefiting from the excellent color-tunable CPOA emission, we take one step further to indicate the feasibility of (R, R)-DAACH and (S, S)-DAACH powders in fabricating multicolor CPL displays and visual UV color chart under ambient conditions (Fig. 4b)[35]. As designed, the "C" and "OO" characters in "COO" pattern were filled with (S, S)-DAACH and (R, R)-DAACH powders, respectively. With the excitation wavelength changing from 254 to 365 nm, the OA images of "COO" were correspondingly changed from blue to yellowish-green, demonstrating the success in achieving multicolor CPOA displays (Fig. 4b, c, top). More interestingly, because of the outstanding CPL feature of (S, S)-DAACH and (R, R)-DAACH powders, the multicolor CPL displays were facilely obtained (black for positive CPL and red for negative CPL) (Fig. 4c, bottom). More strikingly, since the color-tunable CPOA images upon excitation by invisible UV light spanning from 240 to 360 nm with a step of 40 nm can be easily recognized by the

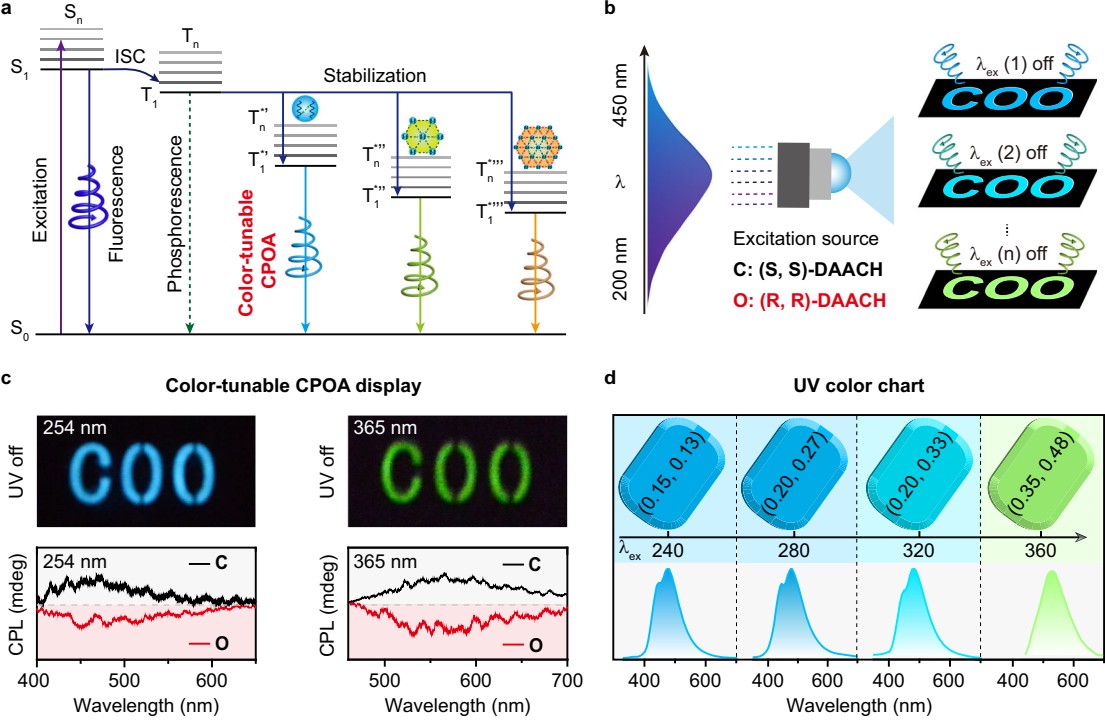

**Fig. 4 Mechanism and applications of the color-tunable CPOA molecules. a** Plausible energy transfer processes for color-tunable CPOA. **b** Schematic drawing of multicolor CPOA displays by varying the excitation wavelength of UV light. **c** Color-tunable CPOA display of COO pattern using (S, S)-DAACH to fabricate "C" and (R, R)-DAACH for "OO" and the corresponding CPL curves of "C" (black) and "O" (red) in the chiral-featured pattern upon 254 (left) and 365 (right) nm UV light excitation, respectively. **d** Excitation-dependent color chart of CPOA spectra with the corresponding CIE values demonstrating the possibility of (R, R)-DAACH powder in serving as sensing medium for visual RTP detection of particular UV light wavelengths.

naked eyes, these chiral cluster crystals may hold substantial promise for advanced visualization of UV sensing (Fig. 4d).

## Discussion

In summary, we have succeeded in achieving color-tunable CPOA from a single-component chiral cluster crystal through chiral clusterization. The direct linkage between the chirality center of *trans*-1, 2-diamidocyclohexane, and succinic acid containing multiple heteroatoms can not only yield efficient CPL emission and diverse CTE emission species but also boost efficient ISC and strong non-covalent intermolecular interactions to promote the generation of triplet excitons and suppression of non-radiative decay for efficient RTP. More importantly, owing to the spectacular chiral clusterization effects, dynamically selective response to the excitation wavelength for color-tunable CPOA emission was observed in the single-component molecular clusters. With the changing of excitation wavelength from 240 to 360 nm, the two enantiomers of (S, S)-DAACH and (R, R)-DAACH exhibited excellent color-tunable CPOA spanning from blue (~470 nm) to yellowish-green (~530 nm) with $|g_{lum}|$ values higher than $2.3 \times 10^{-3}$ and lifetime up to 587 ms. Taking advantage of the spectacular excitation-dependent color-tunable CPOA molecules, multicolor CPL displays and visual UV color charts were successfully constructed. The realization of multicolor-tunable CPOA emission from a single-component chiral cluster crystal through a one-stone-two-birds approach could afford fundamental and vital design clues for the exploration of smart multicolor CPL and RTP functional materials, shedding important light on the construction of multifunctional and high-performance new-concept organic molecules for advanced optoelectronic applications.

## Methods

Unless otherwise noted, all reagents were purchased from Aldrich and Acros and used without additional purification. The molecular structures of the as-synthesized phosphorescent molecules were characterized by single-crystal X-ray diffraction, NMR spectroscopy, HRMS, and MALDI-TOF MS (see the Supplementary Information).

The steady-state UV absorption data were collected on a Jasco V-750 spectrophotometer. The intrinsic circularly polarized luminescence (CPL) spectra were investigated using a JASCO CPL-300 spectrometer. The circular dichroism (CD) spectra were measured on a JASCO J-810 circular dichroism spectrometer. The fluorescence spectra, phosphorescence spectra, kinetic measurements, lifetimes of the room-temperature phosphorescence (RTP) were measured using an Edinburgh FLS980 fluorescence spectrophotometer equipped with a xenon arc lamp (Xe900), laser, and a microsecond (μs) flash-lamp (uF900), respectively. All the data of these single-crystal structures were collected on a Bruker SMART APEX (II)-CCD at 100 K. The photos and supporting videos were recorded by Nikon D90. The TD-DFT calculations were performed with the Gaussian 09 program. All the computational models were built from single-crystal structures without further geometry optimization. The excitation energy of the *n*th singlet state ($S_n$) and the *n*-th triplet state ($T_n$) states were calculated at the TD-DFT method of B3LYP/6-31 G(d, p) level based on the monomer and selected aggregates extracted from the single-crystal. Frontier molecular orbital distributions were calculated based on the monomer and selected aggregates extracted from the single-crystal at M06-2X/cc-pVTZ.

## Data availability

The data that support the findings of this study are available from the corresponding author upon request. Crystallographic data generated in this study for (R,R)-DAACH has been deposited in The Cambridge Crystallographic Data Centre under accession code CCDC 2083718.

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

## Acknowledgements

This study was supported in part by the National Natural Science Foundation of China (62075102 awarded to HL, 22075149 awarded to YT, 21772095 awarded to CZ, 61875090, and 91833306 awarded to RC), the Jiangsu Specially-Appointed Professor Plan (awarded to YT), the Six Talent Plan of Jiangsu Province (XCL-049 awarded to YT), Key giant project of Jiangsu Educational Committee (19KJA180005 awarded to RC), the fifth 333 project of Jiangsu Province of China (BRA2019080 awarded to RC), 1311 Talents Program of Nanjing University of Posts and Telecommunications (Dingshan awarded to YT), Natural Science Fund for Colleges and Universities in Jiangsu Province (20KJB430001 awarded to HL), China Postdoctoral Science Foundation-funded project (2018M642284 awarded to HL), and Nanjing University of Posts and Telecommunications Start-up Fund (NUPTSF) (NY220151 awarded to GX, NY219007 awarded to YT, NY219160 awarded to PL, and NY217140 awarded to HL).

## Author contributions

HL, YT, WH, and RC conceived and designed the experiments. HL, JW, HL, and GX measured and analyzed the photophysical properties. ZW, CZ, and PL performed the computational calculations. HL, JG, and FH fabricated the applications. HL, YT, WH, and RC wrote the manuscript and all authors contributed to the data analysis.

## Competing interests

The authors declare no competing interests.
