## [Peer Review File · Nature Communications]

REVIEWER COMMENTS

Reviewer #1 (Remarks to the Author):

Comments on NCOMMS-21-31539:

In this MS, circularly polarized organic afterglow from single component DAACH was reported, through the integration of CTE and chiral moieties. Generally, I think the topic and strategy of this work is simple yet interesting. Also, the spectroscopic characterizations are deserved to be praised. I have the following concerns:

1. The design rationale for single component CPOA: here the authors conjugated the chiral center (trans-1,2-diamidocyclohexane) with the TSC center. I wonder whether such design rationale can be extended with other chiral center or other emission centers? If yes, the authors should provide several other successful examples.
2. For theoretically probing the luminescence mechanism, the authors presented the data of monomer, dimer, and tetramer. Also, the authors confirmed that the molecular packing and intermolecular electronic communications is of vital importance. Therefore, I think the theoretical calculation of cluster should be more representative.
3. For CPL, $|g_{lum}|$ is an important parameter. How about the $|g_{lum}|$ of this work when compared with the existing ones? What's the decisive parameter that controls $|g_{lum}|$?
4. As shown in Figure 2f, DAACH exhibited photo irradiation-enhanced RTP, but no experimental or theoretical explanations were given.
5. The explanations about the color-tunable emission seems plausible. How about the Ex-dependent lifetime?
6. For the application of the color-tunable CPOA: I am not sure what's the role of circularly polarized emission. In fact, only the afterglow emission was explored.

Reviewer #2 (Remarks to the Author):

The manuscript by Li and co-authors reported an interesting strategy to realize color-tunable circularly polarized organic afterglow in single component by integrating clusteroluminescence into chiral crystal system. By virtue of the intrinsic excitation-dependent emission of clusters, tunable organic afterglow emissions from blue to yellowish green with circularly polarized property were observed at room temperature. The experimental and theoretical dates are solid. However, I have some concerns about the CPL data shown in the present manuscript. CPL spectrum measured at CPL200 instrument always offers a blend of fluorescence and phosphorescence components. It is impossible to selectively collect the phosphorescence component of the CPL emission by delay setting as traditional photoluminescence instrument. Thus, the CPL data shown in Figure 2b should consist of both fluorescence and afterglow components. Based on such consideration, the DC panel in Figure 2b should be consistent with the steady-state photoluminescence spectra shown in Figure S17 rather than the afterglow spectra shown in Figure S9. However, I found the DC panel in Figure 2b consist of only afterglow component. How did authors realize this? It seems strange. The authors should give a clear description of CPL instrument, parameter setting and measurement procedure. I strongly believe the CPL spectra in Figure 2b consist of dominant circularly polarized fluorescence and tiny amount of circularly polarized afterglow. These concerns needed to be carefully addressed.

Other comments:

1. Crystals always have large birefringence and line dichroism effect, which could potentially result in artificial CPL signal. Is the powder used for the CPL measurement crystal? What is the size of the crystal powder? I have a lot of CPL measurement experiences. I recommend the authors to measure the CPL of powder in KBr pellet to get more reliable CPL signal. Actually, the CPL signal excited by 240 nm shown in Figure 2b is so weak that it is difficult to tell whether it is a real signal or a noise.
2. What is the monitored wavelength for the time-resolved decay profiles in Figure S11 and S23?

3. How did the authors prepare the (R,R)-DAACH in ethanol solutions? Disperse the powder into ethanol? Does the (R,R)-DAACH in ethanol solutions still show afterglow?
4. Why does the SSPL intensity but not the afterglow increase with the increase of excitation power in Figure S15 c and d?
5. The crystal showed interesting excitation-dependent afterglow, but in the steady-state photoluminescence spectra, no excitation-dependence was observed, which means the fluorescence component showed no excitation-dependence. Why?
6. When people measure and discuss CPL spectra, CD spectra are always simultaneously provided to give more fruitful information. Since (R,R)-DAACH in ethanol solutions showed observable absorption spectra, CD spectra should be also provided.

Reviewer comments:

Reviewer: #1

Comment 1: *In this MS, circularly polarized organic afterglow from single component DAACH was reported, through the integration of CTE and chiral moieties. Generally, I think the topic and strategy of this work is simple yet interesting. Also, the spectroscopic characterizations are deserved to be praised. I have the following concerns:*

Author reply: Thanks for the referee's professional comments and kind recommendation of our work. We have carefully addressed the points raised by the referee.

(1) The design rationale for single component CPOA: here the authors conjugated the chiral center (trans-1, 2-diamidocyclohexane) with the TSC center. I wonder whether such design rationale can be extended with other chiral center or other emission centers? If yes, the authors should provide several other successful examples.

Author reply: We really appreciate the professional comments and thoughtful suggestions. Indeed, this chiral clusterization strategy can be extended to other single-component color-tunable CPOA systems with different chiral centers and/or emission centers, confirming the universality of our strategy. In the revised manuscript, other two pairs of enantiomers of **(R, R)/(S, S)-DAPCH** and **L/D-serine** were explored (**Supplementary Figs. 39-40**). As expected, **(R, R)/(S, S)-DAPCH** also demonstrate superb excitation-dependent CPOA properties, showing tunable emission color ranging from 410 nm to 530 nm (**Supplementary Fig. 39**). And, the single-component color-tunable CPOA can be also achieved in **L/D-serine**, exhibiting tunable chiral afterglow emissions spanning from deep blue to green (**Supplementary Fig. 40**). We have involved corresponding discussions in the revised manuscript and Supplementary Information. Thanks a lot.

Added text and figure:

Page 6 of the revised manuscript, left column, lines 28-36 and Supplementary Figs. 39-40:

To testify the universality of this chiral clusterization strategy in constructing single-component

color-tunable CPOA materials, other two pairs of enantiomers of **(R, R)/(S, S)-DAPCH** and **L/D-serine** were explored. Indeed, **(R, R)/(S, S)-DAPCH** demonstrate superb excitation-dependent CPOA properties, showing tunable emission color ranging from 410 to 530 nm (**Supplementary Fig. 39**). And, the single-component color-tunable CPOA can be also achieved in **L/D-serine**, exhibiting tunable chiral afterglow emissions spanning from deep blue to green (**Supplementary Fig. 40**).

Figure S39. (a) Chemical structures of **(R, R)/(S, S)-DAPCH**. (b) Afterglow spectra and (c) corresponding CPL properties of **(R, R)/(S, S)-DAPCH** powders upon excitation at different wavelength under ambient conditions.

Figure S40. (a) Chemical structures of L/D-Serine. (b) Afterglow spectra and (c) corresponding CPL properties of L/D-Serine powders upon excitation at different wavelength under ambient conditions.

(2) For theoretically probing the luminescence mechanism, the authors presented the data of monomer, dimer, and tetramer. Also, the authors confirmed that the molecular packing and intermolecular electronic communications is of vital importance. Therefore, I think the theoretical calculation of cluster should be more representative.

Author reply: Many thanks for the professional suggestions. The theoretical calculation of cluster has been updated and the corresponding additional discussions have been added in the revised manuscript and Supplementary Information.

Added text and figures:

Page 5 of the revised manuscript, right column, lines 24-26 and Supplementary Fig. 37:

Furthermore, with the increment of the aggregate size, more obvious through-space delocalization and decreased energy bandgaps were observed (**Fig. 3e** and **Supplementary Fig. 37**).

Figure S37. (a) Frontier molecular orbital distributions for the selected hexamer and octamer and (b) calculated energy bandgaps for the selected aggregates extracted from (**R, R**)-DAACH single crystal.

(3) For CPL, $|g_{lum}|$ is an important parameter. How about the $|g_{lum}|$ of this work when compared with the exiting ones? What's the decisive parameter that controls $|g_{lum}|$?

Author reply: We appreciate the professional question raised by the referee. The $|g_{lum}|$ s of our work are moderate when compared with the recently published dissymmetry factor of the afterglow materials showing $|g_{lum}|$ s ranging from 1.46×10^{-3} to 2.0×10^{-2} (**Supplementary Fig. 14**). Theoretically, g_{lum} can be calculated from the following equation:

$$g_{lum} = \frac{4|m|\cos\theta}{|\mu|}$$

where $|m|$ and $|\mu|$ are the magnitudes of magnetic and electric transition dipole moments vectors, respectively, and the θ is the angle between these two dipole moments. Therefore, large $|\mu|$ means small g_{lum} , while large $|m|$ will result in a high g_{lum} value (*Angew. Chem. Int. Ed.* **2019**, *58*, 7013-7019; *Angew. Chem. Int. Ed.* **2019**, *58*, 4978-4982). We have involved these discussions in the revised manuscript and Supplementary Information. Thanks again.

Added text and figure:

Page 2 of the revised manuscript, right column, lines 6-8:

Excitation-dependent color-tunable CPOA is successfully realized in single-component chiral

cluster crystal of **(R, R)-DAACH** and **(S, S)-DAACH**, showing long-lived emissions ranging from blue to yellowish green with moderate $|g_{lum}|$ over 2.3×10^{-3} under the excitation by 240, 340 and 360 nm, respectively (**Fig. 2a-b** and **Supplementary Fig. 14**).

Page S12 of the revised Supplementary Information:

Experimentally, the dissymmetry factor (g_{lum}) can be calculated from following equation: $g_{lum} = 2(I_L - I_R) / (I_L + I_R)$. Theoretically, g_{lum} is defined as:

$$g_{lum} = \frac{4|m|\cos\theta}{|\mu|}$$

where $|m|$ and $|\mu|$ are the magnitudes of magnetic and electric transition dipole moments vectors, respectively, and the θ is the angle between these two dipole moments. Therefore, large $|\mu|$ means small g_{lum} , while large $|m|$ will result in a high g_{lum} value.

Figure S14. Reported chiral organic afterglow materials with corresponding lifetimes (blue) and g_{lum} (red) under ambient conditions⁵⁻¹⁰.

(4) As shown in Figure 2f, DAACH exhibited photo irradiation-enhanced RTP, but no experimental or theoretical explanations were given.

Author reply: we appreciate the point raised by the referee and many thanks for the careful review. Actually, according to the previous reports and our understanding, the slightly enhanced RTP intensity upon the increased photo irradiation time is not a photo irradiation-enhanced RTP and this is a common phenomenon for organic afterglow materials (*Nat. Mater.* **2015**, *14*, 685-690; *Nat. Photonics*, **2019**, *13*, 406-411). This phenomenon is due to the fact that organic afterglow emission needs more time to achieve the stable state of triplet excitons for emission owing to the relatively slow and weak intersystem crossing rates to populate the triplet excited state. We have involved the possible explanation in the revised manuscript. Thanks a lot.

Added text:

Page 3 of the revised manuscript, right column, lines 2-7:

Possibly due to the relatively slow and weak intersystem crossing rates, it needs considerable time to achieve the stable state of triplet excitons for afterglow emission, resulting in the slightly enhanced RTP with the increase of photo irradiation time (**Fig. 2f** and **Supplementary Fig. 23**).

(5) The explanations about the color-tunable emission seems plausible. How about the Excitation-dependent lifetime?

Author reply: We appreciate the question raised by the referee. Indeed, the excitation-dependent lifetimes were also found in chiral clusters of (R, R)/(S, S)-DAACH (**Supplementary Figs. 20-21**). These variations in lifetime further confirmed the presence of disparate emissive species. To make this excitation-dependent lifetime more intuitive to the readers, the **Supplementary Figs. 20-21** had been updated. Many thanks.

Updated text and figures:

Page 3 of the revised manuscript, left column, lines 8-10:

These distinct variations in lifetime further confirmed the presence of disparate emissive species.

Figure S20. (a) Afterglow decay profiles and (b) corresponding lifetimes of the main emission bands of (R, R)-DAACH powder upon excitation at different wavelength under ambient conditions.

Figure S21. (a) Afterglow decay profiles and (b) corresponding lifetimes of the main emission bands of (S, S)-DAACH powder upon excitation at different wavelength under ambient conditions.

(6) For the application of the color-tunable CPOA: I am not sure what's the role of circularly polarized emission. In fact, only the afterglow emission was explored.

Author reply: We appreciate the concern raised by the referee. As to the role of circularly

polarized emission in the application, we think that the chirality features of **(S, S)-DAACH** and **(R, R)-DAACH** render the multicolor afterglow displays with different circularly polarized emission features. As shown in **Fig. 4c**, although quite similar afterglow emission can be found in **(S, S)-DAACH** and **(R, R)-DAACH** powders, the inherent difference of chirality renders this pair of enantiomers with different circularly polarized emission, showing a negative CPL signal for "C", and a positive CPL signal of "OO". We indeed know that this is a very primary attempt to develop multicolor CPOA displays, but we believe that this multicolor CPOA material system may have great application potential in chiral encryption and 3D displays. To make this CPL properties of displayed patterns more intuitive to the reader, we have updated Fig. 4c in the revised manuscript. We hope these descriptions can clarify the concern raised by the referee. Many thanks.

Figure 4c. Color-tunable CPOA display of COO pattern using **(S, S)-DAACH** to fabricate "C" and **(R, R)-DAACH** for "OO" and the corresponding CPL curves of "C" (black) and "O" (red) in the chiral-featured pattern upon 254 (left) and 365 (right) nm UV light excitation, respectively.

Reviewer: #2

Comment2: *The manuscript by Li and co-authors reported an interesting strategy to realize color-tunable circularly polarized organic afterglow in single component by integrating clusteroluminescence into chiral crystal system. By virtue of the intrinsic excitation-dependent emission of clusters, tunable organic afterglow emissions from blue to yellowish green with circularly polarized property were observed at room temperature. The experimental and theoretical dates are solid.*

Author reply: Thanks for the professional comments and recommendation of our work.

Comment 2.1: *However, I have some concerns about the CPL data shown in the present manuscript. CPL spectrum measured at CPL200 instrument always offers a blend of fluorescence and phosphorescence components. It is impossible to selectively collect the phosphorescence component of the CPL emission by delay setting as traditional photoluminescence instrument. Thus, the CPL data shown in Figure 2b should consist of both fluorescence and afterglow components. Based on such consideration, the DC panel in Figure 2b should be consistent with the steady-state photoluminescence spectra shown in Figure S17 rather than the afterglow spectra shown in Figure S9. However, I found the DC panel in Figure 2b consist of only afterglow component. How did authors realize this? It seems strange. The authors should give a clear description of CPL instrument, parameter setting and measurement procedure. I strongly believe the CPL spectra in Figure 2b consist of dominant circularly polarized fluorescence and tiny amount of circularly polarized afterglow. These concerns needed to be carefully addressed.*

Author reply: We appreciate the professional questions and constructive suggestions raised by the referee. The intrinsic circularly polarized luminescence (CPL) spectra were investigated using a JASCO CPL-300 spectrometer. To make the CPL measurement more intuitive to the readers, we have supplied the detailed determination methods, and the corresponding description of CPL instrument, parameter setting and measurement procedure according to the referee's suggestions. We totally agree with the referee that we can not obtain the pure phosphorescence component of CPL emission due to the lack of time-resolved technology in

CPL-300 instrument and the CPL emissions measured by CPL-300 instrument should consist of both fluorescence and phosphorescence components. However, in light of the large Stokes-shift of the fluorescence and phosphorescence in organic afterglow materials, the CPL spectra can be split into fluorescence and phosphorescence parts through using the short-pass and long-pass filters to achieve the fluorescence and phosphorescence signal of CPL emission (*Chem.-Eur. J.* **2018**, *24*, 17444-17448). Experimentally, when we measured the fluorescence component of CPL emission, the short-pass filter was used to eliminate the influence of the phosphorescence emission on the fluorescence signal of CPL emission; and, the long-pass filter was employed to suppress the influence of the fluorescence emission on the phosphorescence signal of CPL emission. We do know this method is not accurate enough to measure the true fluorescence and phosphorescence components of CPL emission, but we think this method should be acceptable and has been used in some recent publications (*J. Am. Chem. Soc.* **2021**, *143*, 18527–18535).

Specifically, because of the very small Stokes-shift between the fluorescence and phosphorescence spectra excited by 240 nm, we can not use the filter to eliminate the influence of fluorescence signal of CPL emission on phosphorescence signal of CPL emission. Therefore, the CPL spectra of SSPL in **Fig. 2b** excited by 240 nm UV light indeed consist of dominant circularly polarized fluorescence and tiny amount of circularly polarized afterglow; thus, the curve (black, DC panel in **Fig. 2b**) excited by 240 nm is similar to that of the SSPL emission excited by 240 nm with main emission peak at ~434 nm (**Supplementary Fig. 26a**), rather than the afterglow emission excited by 240 nm (**Supplementary Fig. 18**). Since the fluorescence component can be effectively eliminated by means of long-pass filter with a cut-on wavelength of 495 nm, for 360 nm excited CPL measurement, the curve (red, DC panel in **Fig. 2b**) is close to that of the afterglow emission excited by 360 nm (**Supplementary Fig. 18**). To further demonstrate the excitation-dependent CPOA properties, CPL properties of the organic afterglow of (**R, R**)-DAACH and (**S, S**)-DAACH powders excited by 340 nm were further performed. As shown in updated **Fig. 2b**, the curve (blue, DC panel in **Fig. 2b**) excited by 340 nm is also identical to that of the afterglow emission excited by 340 nm (**Supplementary Fig. 18**) and the clearly red-shifted CPOA properties were obviously observed in DC panel when changed

the excitation wavelength from 340 to 360 nm. We hope these explanations can clarify the referee's concerns. Thanks again.

Figure S26a. SSPL spectra of (R, R)-DAACH powder upon excitation at different wavelength under ambient conditions.

Figure S18. Afterglow spectra of (R, R)-DAACH powder upon excitation at different wavelength under ambient conditions.

Added text and updated figure:

Page S12 of the revised Supplementary information:

To individually achieve the fluorescence and phosphorescence signal of CPL emission, the CPL spectra can be split into fluorescence and phosphorescence parts through using the short-pass and long-pass filters owing to the large Stokes-shift of the fluorescence and phosphorescence in organic afterglow materials. Experimentally, when we measured the fluorescence

component of CPL emission, the short-pass filter was used to eliminate the influence of the phosphorescence emission on the fluorescence signal of CPL emission; and, the long-pass filter was employed to suppress the influence of the fluorescence emission on the phosphorescence signal of CPL emission.

Figure 2b. CPL properties of **(R, R)-DAACH** and **(S, S)-DAACH** powders under excitation by 240, 340 and 360 nm, respectively.

(1) Crystals always have large birefringence and line dichroism effect, which could potentially result in artificial CPL signal. Is the powder used for the CPL measurement crystal? What is the size of the crystal powder? I have a lot of CPL measurement experiences. I recommend the authors to measure the CPL of powder in KBr pellet to get more reliable CPL signal. Actually, the CPL signal excited by 240 nm shown in Figure 2b is so weak that it is difficult to tell whether it is a real signal or a noise.

Author reply: Thanks for the professional comments and thoughtful suggestions. We totally agree with the referee that the crystals always have large birefringence and line dichroism effect, leading to artificial CPL signal. To eliminate the interference of large birefringence and line dichroism effect of crystals, the CPL signals in **Fig. 2b** were measured on the basis of the ground **(R, R)-DAACH** and **(S, S)-DAACH** powders with sizes of 4-20 μm . As suggested by the referee, we also measured the CPL properties of **(R, R)-DAACH** and **(S, S)-DAACH** powders in KBr slice with a weight concentration of 50 wt%. Chiral characteristics were also observed

(Supplementary Fig. 15), suggesting that the chirality properties measured in pure (**R, R**)-**DAACH** and (**S, S**)-**DAACH** powders are reliable. Notably, compared to the (**R, R**)-**DAACH** and (**S, S**)-**DAACH** powders, the relatively poor CPL properties of (**R, R**)-**DAACH** and (**S, S**)-**DAACH**-doped (50 wt%) KBr slices may be due to the destructions of the cluster aggregation structure when doping (**R, R**)-**DAACH** and (**S, S**)-**DAACH** powders into KBr. Additionally, we truly understand the concern that concentrated on the CPL signal excited by 240 nm, owing to its relatively large fluctuations in the CPL profiles. To eliminate the influence of noise on the true CPL signal, the superimposed measurements, which have been recognized as an effective method to increase the ratio of signal to noise of CPL signal, were adopted with total numbers of 10~30 cycles when we performed the CPL measurement. Moreover, the CPL curves of (**R, R**)-**DAACH** and (**S, S**)-**DAACH** powders excited by 240 nm demonstrate good configuration symmetry. Therefore, we believe that the CPL signal excited by 240 nm in **Fig. 2b** in our manuscript is true and reliable. We hope these discussions would clarify the referee's concern. Many thanks.

Added text and figures:

Page 2 of the revised manuscript, right column, lines 8-11 and Supplementary Fig. 15:

The chiral characteristics were also observed, when (**R, R**)-**DAACH** and (**S, S**)-**DAACH** powders were dispersed (50 wt%) in potassium bromide slice (**Supplementary Fig. 15**).

Figure S15. CPL properties of the steady-state photoluminescence (SSPL) of (**R, R**)-**DAACH** and (**S, S**)-**DAACH** (50 wt%) dispersed in potassium bromide (KBr) slices when excited by 300 nm

under ambient conditions.

(2) What is the monitored wavelength for the time-resolved decay profiles in Figure S11 and S23?

Author reply: We thank the question raised by the referee. We are very sorry that we forgot to add the monitored wavelength for the time-resolved decay profiles in **Supplementary Figs. 20, 21 and 32**. We have updated the monitored emission wavelengths of **Supplementary Figs. 20, 21 and 32** in the revised manuscript. Thanks again.

Updated figures:

Figure S20. (a) Afterglow decay profiles and (b) corresponding lifetimes of the main emission bands of (R, R)-DAACH powder upon excitation at different wavelength under ambient conditions.

Figure S21. (a) Afterglow decay profiles and (b) corresponding lifetimes of the main emission bands of (S, S)-DAACH powder upon excitation at different wavelength under ambient

conditions.

Figure S32. Time-resolved decay profiles of emission band (434 nm) at varied concentrations in ethanol solution of **(R, R)-DAACH** excited by 300 nm under ambient conditions.

(3) How did the authors prepare the (R, R)-DAACH in ethanol solutions? Disperse the powder into ethanol? Does the (R, R)-DAACH in ethanol solutions still show afterglow?

Author reply: We really appreciate the professional comments and questions. The **(R, R)-DAACH** powder has good solubility in ethanol solvent with solubility up to 40 mg/mL, thus enabling convenient dissolution of **(R, R)-DAACH** in ethanol to prepare **(R, R)-DAACH** solutions with varied concentrations ranging from $\sim 10^{-5}$ to 10^{-2} M. Although the SSPL emission gradually appears and enhances with the increasing concentration of solution, it is still difficult to observe afterglow emission in solution state mainly due to the fast non-radiative decays of the excitons in solution state. The preparation method of **(R, R)-DAACH** solutions has been provided in Supplementary Information. Thanks.

Added text:

Page S13 of the revised Supplementary information:

The solution of **(R, R)-DAACH** with varied concentrations can be facilely prepared by dissolution of **(R, R)-DAACH** powder in ethanol followed by the sonication for 10 min under ambient conditions.

(4) Why does the SSPL intensity but not the afterglow increase with the increase of excitation power in Figure S15 c and d?

Author reply: We thank the referee for the careful review and the thoughtful questions. In **Supplementary Fig. 24c-d**, with the increase of excitation power, both of the SSPL and afterglow intensity increase significantly. But, due to the large difference between SSPL and afterglow intensity, it seems that the afterglow intensity is not increasing with the increase of excitation power when they use the same y axis in **Supplementary Fig. 24c-d**. To make this enchantment of the SSPL and afterglow intensity more intuitive to the reader, we have redrawn **Supplementary Fig. 24c-d** in the revised Supplementary Information. We are sorry for that.

Updated figure:

Figure S24. (a-b) Photoluminescence intensity profiles of 470 nm emission of (a) (R, R)-DAACH and (b) (S, S)-DAACH powders at different excitation power density of 240 nm irradiation. (c-d) SSPL (black) and organic afterglow (red) intensities as a function of the excitation power under ambient conditions.

(5) The crystal showed interesting excitation-dependent afterglow, but in the steady-state photoluminescence spectra, no excitation-dependence was observed, which means the fluorescence component showed no excitation-dependence. Why?

Author reply: We thank the referee for the professional comments and careful review. We double checked the data and found that the SSPL spectra also show an excitation-dependent ratiometric variation between the PL emission peaks (434 nm) and shoulders (500 nm). Thus, the excitation-dependent SSPL emission with varied CIE coordinates were also observed under ambient conditions and at 77 K as shown in **Supplementary Figs. 26-27**. We have updated these discussions in the revised manuscript. Thanks a lot.

Updated text and figures:

Page 4 of the revised manuscript, left column, lines 4-6 and Supplementary Figs. 26-27:

The SSPL of **(R, R)-DAACH** is also excitation-dependent, exhibiting hypochromatic shift for varied colors from sky-blue to blue and tunable quantum yields when the excitation wavelength was changed from 260 to 360 nm.

Figure S26. (a) SSPL spectra, (b) emission intensity ratios between 434 and 500 nm and (c) CIE coordinates of (R, R)-DAACH powder upon excitation at different wavelength under ambient conditions.

Figure S27. (a) SSPL spectra, (b) emission intensity ratios between 434 and 500 nm and (c) CIE coordinates of (R, R)-DAACH powder upon excitation at different wavelength at 77 K.

(6) When people measure and discuss CPL spectra, CD spectra are always simultaneously provided to give more fruitful information. Since (R,R)-DAACH in ethanol solutions showed observable absorption spectra, CD spectra should be also provided.

Author reply: Thank the referee for the professional and constructive suggestion. The CD spectra have been measured and the corresponding discussion and spectra had been provided in the revised manuscript and Supplementary Information. Thanks again.

Added text and figure:

Page 2 of the revised manuscript, right column, lines 11-15 and Supplementary Fig. 16:

The almost mirror circular dichroism signals of (S, S)-DAACH and (R, R)-DAACH ethanol solutions with strong signals at ~205 nm further confirm the successful introduction of chirality in these materials (**Supplementary Fig. 16**).

Figure S16. Circular dichroism (top) and UV-absorption (bottom) spectra of (S, S)-DAACH and (R, R)-DAACH in ethanol solutions (10^{-4} M) under ambient conditions.

A list of changes made:

(1) The list of reported chiral organic afterglow materials with corresponding lifetimes, efficiencies and g_{lum} has been provided in the revised **Supplementary Fig. 14**.

(2) The numbers assigned to references and Supplementary figures have been updated.

- (3) CPL properties of the SSPL of **(R, R)-DAACH** and **(S, S)-DAACH** (50 wt%) doped slice in KBr excited by 300 nm have been measured (**Supplementary Fig. 15**).
- (4) The figures of afterglow decay profiles and corresponding lifetimes of the main emission bands of **(R, R)-DAACH** and **(S, S)-DAACH** powders upon excitation at different wavelength have been updated (**Supplementary Figs. 20-21**).
- (5) **Figure 4c**, **Supplementary Fig. 24c-d** and **Supplementary Figs. 26-27** have been redrawn and updated.
- (6) The monitored wavelength for the time-resolved decay profiles in **Supplementary Figs. 20, 21** and **32** have been labelled.
- (7) Circular dichroism spectra of **(S, S)-DAACH** and **(R, R)-DAACH** in ethanol solutions (10^{-4} M) have been measured and added in Supplementary Information (**Supplementary Fig. 16**).
- (8) Frontier molecular orbital distributions for the selected hexamer and octamer and energy bandgaps for the selected aggregates in **(R, R)-DAACH** single crystal have been calculated and added in Supplementary Information (**Supplementary Fig. 37**).
- (9) Two pairs of enantiomers were explored to demonstrate the universality of the chiral cluster crystal strategy (**Supplementary Figs. 39-40**).
- (10) CPL properties of the organic afterglow of **(R, R)-DAACH** and **(S, S)-DAACH** powders excited by 340 nm have been added in the revised **Fig. 2b**.
- (11) The ^1H , ^{13}C NMR and MALDI-TOF spectra of the newly synthesized enantiomers **(R, R)-DAPCH** and **(S, S)-DAPCH** have been added in Supplementary Information (**Supplementary Figs. 7-12**).

REVIEWERS' COMMENTS

Reviewer #1 (Remarks to the Author):

I think the revisions and responses made by the authors are acceptable. I am pleased to recommend publication.

Reviewer #2 (Remarks to the Author):

The authors have adequately revised their manuscript according to my previous comments and suggestions. The quality of the manuscript has been improved after the revision. I do not have further criticism of the work.

Reviewer comments:

Reviewer: #1

1. I think the reversions and responses made by the authors are acceptable. I am pleased to recommend publication.

Author reply: We appreciate the reviewer's acceptance and recommendation of our work!

Reviewer: #2

1. The authors have adequately revised their manuscript according to my previous comments and suggestions. The quality of the manuscript has been improved after the revision. I do not have further criticism of the work.

Author reply: We are very grateful for the recommendation!